# The Effect of the Type and Amount of Synthetic Fibers on the Effectiveness of Dispersed Reinforcement in Soil-Cements

**DOI:** 10.3390/ma13183917

**Published:** 2020-09-04

**Authors:** Krystian Brasse, Tomasz Tracz, Tomasz Zdeb

**Affiliations:** Chair of Building Materials Engineering, Faculty of Civil Engineering, Cracow University of Technology, Warszawska St. 24, 31-155 Cracow, Poland; tomasz.tracz@pk.edu.pl (T.T.); tomasz.zdeb@pk.edu.pl (T.Z.)

**Keywords:** soil-cement, dispersed reinforcement, polypropylene fibers, bending tests

## Abstract

The paper deals with mechanical properties of soil-cement composites made with non-cohesive soil and reinforced with dispersed fibers. The research was carried out on the basis of three soil-cement matrices whose compositions varied in terms of the volumetric fraction of cement paste and the water-cement ratio. Two types of polypropylene fibers were used as dispersed reinforcement: single fibrillated-tapes polypropylene fibers (SFPF) and bundles of coiled fibrillated-tapes polypropylene fibers (BCFPF). The fibers varied in terms of their length and mass fraction. The objective of the study was to assess the effect of the addition of fibers to soil-cement composites on their flexural tensile strength and on their behavior in the post-critical state. The studies were carried out after 28 days of curing. Bending tests were carried out to determine post-critical stress values σ_CMODi_, stress values at which the matrix is destroyed (limit of proportionality) σ_LOP_, maximum stress values transferred by the fibers σ_MOR_ (modulus of rupture), and total fracture energy G_f,tot_ as well as compressive strength. The test results obtained, and their analysis, indicate the significant impact of the dispersed reinforcement used on the performance of such composites during bending.

## 1. Introduction

Ground improvement is a long-established procedure in geotechnical engineering. The rapid development of this technology over the years has resulted in many methods of improving weak ground. The need to classify these methods was equally quickly recognized. In the early 1980s, Mitchell [1] classified them in 6 categories: in situ deep densification of non-cohesive soils; consolidation by preloading with/without vertical drains; injections and grouting; stabilization using admixtures; thermal stabilization; and reinforcement of soil. In 2000, Terashi and Juran [2] supplemented this classification with another category: soil replacement. The TC17 Ground Improvement Technical Committee, acting within the International Society for Soil Mechanics and Geotechnical Engineering, in its report of 2009 [3], classified ground improvement methods in 5 categories, depending on whether foreign substances are introduced into the soil or not. Another factor determining the allocation to a given category is the type of improved soil: cohesive or non-cohesive. In this way, the following were distinguished: ground improvement without admixtures in non-cohesive soils or filling materials, ground improvement without admixtures in cohesive soils, ground improvement with admixtures or inclusions, ground improvement with admixtures in the form of injection and soil reinforcement.

Within the category of soil reinforcement, the TC17 Technical Committee [3] distinguishes the method: Geosynthetically or mechanically stabilized soil and characterizes it as the use of tensile strength of various materials to increase the shear strength of soil as well as the stability of roads, foundations, embankments, slopes, or retaining walls. This method covers a wide range of materials that Chu et al. [4] assign to the site of reinforcement application. Among the types of reinforcement, there are: Steel strips, meshes or rods, a range of polymer materials, and micro elements such as natural, glass, or carbon fibers. It was also emphasized that soil reinforcement has become a very attractive alternative to retaining walls or steep slope stabilization because of its advantages in terms of reliability, flexibility, cost, and aesthetics.

Among many methods of ground improvement, the technique of deep mixing of soil with mineral binders in situ also draws attention, as it allows for direct modification of soil properties without the necessity to bring it to the surface. The procedure of introducing a mineral binder into the ground and their deep mixing results in the creation of a composite called soil-cement. It is currently considered a material with many applications. These include anti-filtration barriers, e.g., at flood embankments, protection of the soil against liquefaction, stabilization of embankments, strengthening of embankments, sealing of the ground or even foundation of buildings. Deep excavations and retaining structures are currently more and more frequent as a field of use for this material. However, such structures are primarily exposed to horizontal loads, e.g., in the form of ground pressure or hydrostatic ground water pressure, which generate bending moments. Soil-cement, on the other hand, is a material that tends to be brittle, so its plastic deformation is relatively low. Such characteristics are not advantageous when the stabilized soil is subject to lateral pressure, as it loses its stability due to cracking. Therefore, there is a need for reinforcement, mainly due to the tensile stress. So far, the desired reinforcement is provided by means of steel sections, which are inserted into the wall before the setting of the soil-cement mix. This solution generates additional labor and costs. Therefore, there was a need to reduce the spacing of such sections, or for smaller constructions—to abandon this type of reinforcement entirely and use an alternative solution. An innovative material modification is the introduction of dispersed reinforcement in the form of synthetic fibers into the mineral matrix of soil-cement. The composite created in this way is called fiber-reinforced soil-cement. The purpose of introducing fibers into the soil-cement is to increase its flexural tensile strength as well as to enable the soil-cement structure to retain working properties after cracking in the post-critical state.

After literature review, it can be stated that a modern method of soil reinforcement is the use of continuous, flat geosynthetics: geotextiles, geonets, geofabrics, geomembranes. Their detailed characteristics and wide application in soil structures can be found in the study by the team of the Road and Bridge Research Institute [5]. However, in his doctoral dissertation of 2005, Li [6] points out that this type of reinforcement provides an increase in soil strength in one specific direction, and at its contact with the soil, a plane of reduced (in comparison with unreinforced soil) shear strength is formed. In addition, the use of geosynthetics requires a suitably designed anchorage of sufficient length to ensure the required tensile strength. Another way of reinforcement is short and randomly oriented fibers in the ground. Li emphasizes that fibers evenly mixed with the ground can provide an isotropic increase in the strength of the composite without introducing planes of reduced shear strength. In his opinion, this type of reinforcement also does not require design considerations regarding anchoring. Madej [7], in turn, characterizes the behavior of fibers in the ground, comparing it to that of plant roots. Thanks to their presence, the shear stresses formed in the ground are distributed through inclusions with relatively high tensile strength. This behavior ensures an increase in the stability of the soil as well as its mechanical properties.

Hejazi et al. [8] compiled a broad review of the literature regarding the properties, applications, and effects of fibers in reinforced soils. At the beginning, the authors emphasize that soil reinforcement with fibers requires precise analysis in terms of optimizing their properties, diameter, length, surface texture, or reinforcement mechanism. Soil reinforcement can be divided into two categories, depending on its modulus of elasticity and, consequently, the performance characteristics in the ground medium. We distinguish between high modulus of elasticity (rigid) and low modulus of elasticity (ductile). The first group (e.g., steel strips, glass or basalt fibers) is primarily intended to strengthen the soil and limit its internal and boundary deformations. The second group includes natural and synthetic fibers and geosynthetics, which affect the achievement of a certain level of soil reinforcement, but more importantly, they provide operational properties in post-critical stress areas. The development of plastics production technology has led to a growing interest in the modern industry in fibers made of plastics. Among synthetic fibers, one can mention: polypropylene PP, polyester, most frequently PET, polyethylene PE, glass, nylon, or polyvinyl alcohol PVA fibers.

Polypropylene fibers are one of the most frequently used fibers in engineering practice. They increase the strength properties of e.g., expansive soil (clay), reduce its shrinkage and swelling, as demonstrated by Soğancı [9]. Polypropylene fiber-reinforced soils (e.g., sludge sand) are also characterized by greater cohesion with a slight decrease in the angle of internal friction. PP fiber reinforcement can also provide higher residual strength. Polypropylene fibers are mainly used for surface ground improvement. Field studies presented by Santoni and Webster [10] confirmed the effectiveness of their application in reinforcing sandy soils of airports and roads. However, it is necessary to cover such surfaces with a layer of binder to prevent the fibers from being pulled out by movement. Consoli et al. [11] examined the response to the distributed load of the layer of compacted sandy soil—reinforced with polypropylene fibers and unreinforced. The layer of soil reinforced with fibers was characterized by higher stiffness with an increase in compaction in comparison to the unreinforced layer. At the same time, apart from field tests, laboratory tests of triaxial compression of specimens with and without polypropylene fibers were conducted. In the case of specimens with fibers, significant strength retention was observed until the end of the tests when axial strain exceeded 20%. On the other hand, specimens not reinforced with fibers showed almost perfect plasticity at high strains. The results allow to search for potential applications of polypropylene fiber reinforcement in shallow foundations, embankments on soft soils, and other earthworks exposed to excessive strains. Polypropylene fibers are hydrophobic and resistant to the corrosive properties of alkali, chlorides, as well as biological degradation. Another advantage is their relatively low cost. Table 1 shows selected properties of polypropylene fibers used for ground improvement.

## 2. Study Objective

The purpose of the conducted study was to assess the effectiveness of improvement of the soil-cement matrix with dispersed reinforcement in the form of polypropylene fibers. The choice of this type of fibers is motivated by their durability, low cost, and their ability to work in the post-critical condition. The obtained solution is to increase the functionality of civil structures made of soil-cement with dispersed reinforcement. An increase in the flexural tensile strength of the composites in question was expected, with a simultaneous, pseudo-plastic character of their destruction, which enables further operation of the composites tested in the post-critical condition. The scope of research was therefore designed to help optimize the composition of the composites to achieve the required properties.

## 3. Materials and Methods

The research program included producing a series of fiber-reinforced soil-cements, differentiated in terms of type and amount of added fibers. For the study, three matrices with different volumes of dosed cement paste V_z_ and different values of the w/c ratio were selected. The composites were made of non-cohesive soil, tested and designated as medium sand. Table 2 presents the basic properties of this soil.

Soil-cement matrices were produced as a result of introduction of CEM II/B-S 32.5 R cement paste into the native soil characterized above. Its properties are presented in Table 3.

The basic criterion for the selection of three soil-cement compositions for the study was the cement content at the level of approx. 200, 300, and 400 kg/m^3^. The adopted range of cement content ensures the variation in soil-cements mechanical properties and corresponds to the amounts used in geotechnical projects. At the same time, these compositions were to ensure that materials could be incorporated into the structure using methods applicable on a technical scale. They were, therefore, designed to have a suitable liquidity after mixing. The mixes were required to be able to self-compact without separating the components. These requirements conditioned the application of an appropriate amount of cement paste and the adopted w/c ratio. The method of design and the effect of composition variability on the properties of mixes in a wide range were described in [12]. The liquidity of all mixes was determined on a vibrating table used for testing mortars in accordance with the requirements of the standard PN-EN 1015-3: 2000/A2 [13]. The measure of consistency was, on the one hand, quantifiable—the diameter of flow, on the other hand, qualitative—the observation of the behavior of the mix (homogeneity of flow, tendency to separate water, lack of sedimentation of individual components). Thus, the selected mixes achieved a flow value between 180 and 220 mm. The compositions and basic mechanical properties of the selected soil-cement matrices are presented in Table 4.

Two types of polypropylene fibers were analyzed for behavior together with these matrices: single fibrillated-tapes polypropylene fibers (SFPF) and bundles of coiled fibrillated-tapes polypropylene fibers (BCFPF), which differ in terms of length—60 and 100 mm, for details see Figure 1. The choice of the length of fibers used was based on the research on the interface properties of dispersed reinforcement with the soil-cement matrix, presented in item [14].

The characteristic feature of the polypropylene fibers selected for testing is their structure and their behavior during mixing with the soil-cement matrix. As a result of mixing, a single fibrillated tape stretches transversely, making its structure resembling the mesh presented in Figure 2 visible. This mesh is filled with the soil-cement mix, which increases the contact surface between the fiber and the matrix. In the case of BCFPF fibers, during the mixing process, the spaces between the individual fiber-forming tapes are filled with the soil-cement mix. This increases the adhesion of the fiber to the matrix.

Two levels of fiber dosage were used in fiber-reinforced soil-cement composites: 6 and 10 kg/m^3^. The introduction of fibers and their mixing with the matrix resulted in a slight decrease in the liquidity of the mixes, however, the consistency obtained still allowed for the incorporation of the mix by methods used on a technical scale. Immediately after the components were mixed, cuboid specimens of 150 mm × 150 mm × 600 mm were formed. The dimensions of the specimens in molds ensured that they were representative in relation to the geometry of the fibers used. After mold release, the specimens were tightly wrapped with foil to prevent moisture exchange with the environment and stored in laboratory conditions at a temperature of 20 ± 2 °C (constant mass of specimen elements). The mechanical properties of the composites in question were tested after 28 days of curing.

The methodology of testing the effectiveness of dispersed reinforcement in fiber-reinforced soil-cement has not been standardized yet. Jamsawang et al. [15,16] used the guidelines contained in ASTM C1609 [17] for flexural tensile strength tests of reinforced concrete with dispersed reinforcement. One of the documents, which is dedicated to the testing of the tensile strength at bending of concrete is the standard PN-EN 14651:2007 [18]. The methodology provided in it is often used to test reinforced concrete composites using various types of dispersed reinforcement. Among them: steel fiber-reinforced concrete (SFRC) [19,20,21], self-compacting concrete (SCC) reinforced with synthetic fibers [22,23] or concrete with mixed geo-cement binder reinforced with steel fibers [24]. However, the concrete tensile bending strength test can also be carried out according to the guidelines of the RILEM TC-187 SOC Technical Committee report [25]. Enfedaque et al. [26] applied the methodology included in this document to test the mechanical properties of self-compacting concrete reinforced with polyolefin fibers. The same was done by Sadrmomtazi et al. [27] in the case of heavyweight concrete tests (HWC). Therefore, it was decided to use the guidelines contained in the standard PN-EN 14651 [18] and in the report of the RILEM Technical Committee [25] and to develop on this basis a new methodology for testing the flexural tensile strength of fiber-reinforced soil-cement composites.

The standard PN-EN 14651 [18] makes it possible to determine such parameters as flexural strength, limit of proportionality σ_LOP_ (the stress at which matrix is destroyed), and post-critical stress values σ_CMOD_ corresponding to individual crack mouth opening displacement (CMOD) values of the forming crack. The report of the RILEM Technical Committee [25] is devoted to testing fracture energy and efficiency of dispersed reinforcement. The use of the guidelines contained in this report is important, due to the fact that soil-cement exhibits low strength, especially under bending. This document is dedicated to testing concrete both with and without fibers added. In the case of tests without fibers, the use of counterweights is recommended to compensate for the additional bending force resulting from the specimen’s own weight.

In order to fully reflect the mode of operation of such fiber-reinforced soil-cements, tensile strength under three-point bending was determined, together with the determination of additional parameter values: the limit of proportionality σ_LOP_, post-critical stress values σ_CMOD_, and total fracture energy for the material G_f,tot_. Furthermore, the analysis of the literature on the efficiency of dispersed reinforcement [28,29,30,31,32] led to its description by determining the parameter σ_MOR_ (modulus of rupture), which is defined as the maximum stress transferred by the dispersed reinforcement after the matrix cracking, i.e., after σ_LOP_. The value of σ_LOP_ was determined as the maximum stress at which the linear relationship between stress and crack opening is still observed. The total fracture energy G_f,tot_ was determined as a surface area under the force-deflection curve of 10 mm and referred to the cross-sectional area of the specimen. During the constant increase of the beam deflection during the test, the value of the acting force P, the opening of the crack mouth opening displacement CMOD (Figure 3) and the deflection of beam d (Figure 4) were recorded simultaneously. The standard PN-EN 14651 [18], in the case of fiber-reinforced concrete testing, suggests the completion of measurements at CMOD = 4 mm. However, due to the recorded further increase in the post-critical stress in the case of fiber-reinforced soil-cement, the tests were continued up to CMOD = 10.0 mm. It was only then that a decrease in the post-critical stress was observed. For this reason, the values of the post-critical stresses suggested by PN-EN 14651 from σ_CMOD_0.5_ to σ_CMOD_3.5_ were modified in relation to the original recommendations and recorded at CMOD = 0.5, 2.5, 5.0, 7.5, 10.0 mm. At the same time, the modification of the measurement range of the CMOD allowed to determine the σ_MOR_ parameter. Due to the increase time of crack opening measurement, it was decided to also modify the rate of displacement increase during the test. By the standard PN-EN 14651 [18], the rate of specimen loading is controlled by crack opening and amounts to 0.08 mm/min. In the adopted methodology the specimens were loaded at 0.2 mm/min up to CMOD = 0.3 mm, while above this crack mouth opening displacement value, the rate was increased to 3.0 mm/min. This modification did not affect the values of the parameters recorded, but resulted in a significant shortening of testing time for a single specimen. According to the standard PN-EN 14651, in order to force crack propagation, notches of 25 mm deep and up to 2 mm wide were made in the middle of the beams’ length, while the spacing of the supports was 450 mm.

In order to compensate for the additional bending force resulting from the specimen’s own weight, a frame was made for testing, on which the counterweights were placed at an appropriate distance from the axis of the specimen. At the same time, the frame made it possible to install a sensor used to record the beam deflection without the deformation of the specimen on the supports (pressing the shafts of the supports into the specimen and possible displacement of the supports during the test). The details regarding the test stand can be seen in Figure 5.

The compressive strength test was carried out on 150 mm × 150 mm × 150 mm cubes in accordance with the standard PN-EN 12390-3: 2019-07 [33].

## 4. Results and Discussion

All the mechanical parameters determined are collected in Table 5, Table 6 and Table 7. Due to the number of tests carried out, the presentation of partial results was omitted, and the presented results were limited to the average values from the three measurements. Due to the achieved effect of increased strength of composites as a result of introduction of fibers (σ_MOR_ > σ_LOP_), the flexural tensile strength f_fm_ is identical to σ_MOR_. Therefore, the f_fm_ values are omitted from the tables below. The obtained efficiency of dispersed reinforcement in fiber-reinforced soil-cement composites is illustrated by the graphs presented in Figure 6, Figure 7 and Figure 8. In the Table 5, Table 6, Table 7 and Table 8 and Figure 6, Figure 7, Figure 8, Figure 9, Figure 10, Figure 11, Figure 12, Figure 13, Figure 14, Figure 15, Figure 16 and Figure 17, the markings containing information about: fiber type/fiber length in mm/amount of fibers in kg/m^3^ e.g., BCFPF/60/10 were used.

### 4.1. Effects of Dispersed Reinforcement on Stress Values σ_LOP_ and σ_MOR_

Figure 9 and Figure 10 show σ_LOP_ and σ_MOR_ values for two types of fibers—BCFPF and SFPF, separately. The presented graphs show the influence of the quantity, length, and type of fibers on the parameters of the assessment of the effectiveness of the dispersed reinforcement in the soil-cement matrix.

#### 4.1.1. Effects of Fibers on σ_LOP_ Stress Values

When analyzing the effect of the presence of dispersed reinforcement on the σ_LOP_ stress values compared to the flexural tensile strength f_fm_ of unreinforced concrete, it can be observed in many cases that there is little or even no visible effect of inclusions. In most combinations of compositions, especially in 400/1.6 and 450/1.2 matrices, the σ_LOP_ stress value is several percent higher than f_fm_ for soil-cement. It is rather difficult to find a clear trend for the influence of fiber type, length or quantity in the composition without a clear correlation.

#### 4.1.2. Effects of Fibers on σ_MOR_ Stress Values

First of all, it should be stated that the σ_MOR_ stress values were higher than the σ_LOP_ stress values for all analyzed composites. Thus, in each combination of compositions, the use of fibers resulted in the reinforcement of these composites.

The change of the fiber dosage level from 6 to 10 kg/m^3^ was accompanied by an increase in the σ_MOR_ stress value, each time. Of course, this increase depended on the type of matrix, type and length of fibers.

Taking into account the influence of fiber length on the obtained σ_MOR_ stress values, it can be concluded that in the case of fiber-reinforced soil-cement with 400/1.6 matrix, i.e., the weakest, the increase in length from 60 to 100 mm resulted in a large increase in the σ_MOR_ stress value, especially at the dosage rate of 6 kg/m^3^. Surprisingly, however, this increase at 10 kg/m^3^ was not as great. In Figure 11, the maximum value of ∆σ_MOR_ equal to 1.4 MPa was marked, which was obtained for 6 kg/m^3^ of BCFPF fibers, and a slight increase of ∆σ_MOR_ equal to 0.3 MPa was recorded for 10 kg/m^3^ of BCFPF fibers.

Similarly, in the case of fiber-reinforced soil-cement with a 450/1.2 matrix, increasing the length from 60 to 100 mm resulted in an increase in the σ_MOR_ stress value, on average by 33%. The maximum value of ∆σ_MOR_ equal to 1.6 MPa was obtained for BCFPF fibers in the amount of 10 kg/m^3^. Whereas, the minimum increase of σ_MOR_ value equal to 0.5 MPa was recorded for SFPF fibers in the amount of 6 kg/m^3^. Both increases in σ_MOR_ are presented in Figure 12.

Different results were obtained for fiber-reinforced soil-cement with the 450/0.8 matrix, which is the strongest. Increase in the length of fibers in most analyzed cases did not significantly affect the σ_MOR_ stress values. For example, in the case of 10 kg/m^3^ of SFPF fibers, an increase in their length even caused a decrease in the σ_MOR_ stress value by 0.9 MPa (see Figure 13). Therefore, in this case, it should be assumed that both 60 and 100 mm-long fibers had sufficient anchorage length in a strong matrix, and the dominant parameter determining the σ_MOR_ stress value was the amount of fibers in the composition of the composite, and thus the amount of active fibers in the cracked section of the specimen.

When analyzing the impact of the fiber type, i.e., BCFPF or SFPF on the σ_MOR_ stress values, it can be said to have little effect on the results obtained. For fiber-reinforced soil-cement with a 400/1.6 matrix, practically no effect of dispersed reinforcement type on the obtained mean values of σ_MOR_ was observed. In the case of composites with a 450/1.2 matrix, the application of SFPF fibers resulted in slightly higher σ_MOR_ stress values in comparison to composites with BCFPF fibers. The opposite trend was observed for composites with a 450/0.8 matrix, where the application of BCFPF fibers resulted in several percent higher σ_MOR_ stress values compared to composites with SFPF fibers.

Since the σ_MOR_ stress value was recorded at different values of CMOD, the Table 8 below shows the effect of the type of dispersed reinforcement on cracks opening, at which the maximum value of the σ_MOR_ post-critical stress was reached.

The analysis of crack opening values at reaching σ_MOR_ indicates that for the tested composites, in most cases, they were recorded at CMOD exceeding 4 mm, which is confirmed by the appropriately adopted in the testing methodology increased range of CMOD in relation to PN-EN 14651:2007 [18]. An exception is made for three cases of fiber-reinforced soil-cement with the addition of BCFPF fibers with the length of 60 mm, in which the σ_MOR_ stresses occurred in the range of CMOD from 3.0 to 3.4 mm. Furthermore, a tendency can be observed that with the same length and amount of dispersed reinforcement dosing, SFPF fiber composites reached maximum σ_MOR_ stress values at higher CMOD values in comparison with BCFPF fiber-reinforced soil-cements. Therefore, it can be presumed that a bundle of coiled fibrillated tapes has a lower deformability than single fibrillated tapes, so that at lower CMOD values, it becomes detached from the matrix and thus the value of the recorded post-critical stress decreases.

It also seems interesting to retain the reinforcement scattered in the weakest matrix, i.e., 400/1.6. Regardless of the type of fiber, for their length of 60 mm, the σ_MOR_ value is always lower, while the CMOD at the MOR point is sometimes more than twice the size of the longer fibers. According to the authors, the problem of adhesion of the inclusion to the matrix is very clearly visible here. The reinforcement made of shorter fibers is characterized by an increased number of active fibers in the resulting crack, which allows to achieve maximum stress values with a relatively small crack opening. However, their anchorage at best of 30 mm in each element is significantly less effective than for 100 mm fibers. Thus, there is a relatively rapid decrease in the recorded post-critical stress. In the case of long fibers, on the other hand, their smaller number in the crack results in reaching σ_MOR_ at a crack opening about twice as large. However, the significantly increased anchorage of the fibers results in an increase in the recorded maximum post-critical stress. It should also be added that with stronger matrices, the relationships observed fade away.

### 4.2. Effects of Dispersed Reinforcement on σ_LOP_/σ_MOR_ Ratio Value

The measure of reinforcement of reinforced concrete is the σ_MOR_/σ_LOP_ ratio. The amplification effect is positive when the σ_MOR_/σ_LOP_ ratio is greater than 1.0. Such a case occurred in all studied fiber-reinforced soil-cement composites. The values of σ_MOR_/σ_LOP_ ratios obtained during the tests are presented in Figure 14 for BCFPF fibers and Figure 15 for SFPF fibers.

When analyzing the effect of the amount of fibers on the values of the σ_MOR_/σ_LOP_ ratios, it can be concluded that an increase in the fiber dosage level from 6 to 10 kg/m^3^ results in a clear increase in the σ_MOR_/σ_LOP_ values of 20–50%, but only in relation to fiber-reinforced soil-cements with a 450/0.8 matrix, which ensures the highest adhesion of fibers. For the other matrices, the effect of the increase in σ_MOR_/σ_LOP_ values resulting from the amount of fibers is not clear.

Taking into account the influence of the fiber type on the obtained values of σ_MOR_/σ_LOP_ ratios, it was observed that for composites with 400/1.6 and 450/1.2 matrices, both BCFPF and SFPF fiber types gave similar results, with 60 and 100 mm-long fibers. In contrast, in the case of the strongest 450/0.8 matrix, it is clear that the use of single 100 mm-long fibrillated tapes SFPF gives better results than bundles of coiled fibrillated tapes BCFPF of the same length. However, if shorter fibers with a length of 60 mm are used, the trend is reversed. In the strongest matrix, BCFPF fibers result in higher values of σ_MOR_/σ_LOP_ ratios than SFPF fibers. Apparently, the increased number of active tape bundles causes destruction of the matrix around the adjacent fiber and thus a decrease in the value of the recorded post-critical stress.

The analysis of the results obtained in the course of the tests also allows us to conclude that, in general, an increase in fiber length from 60 to 100 mm results in an increase in σ_MOR_/σ_LOP_ values. As already mentioned in paragraph 4.1.2., it appears that in matrices with a lower strength, the value of σ_MOR_ stresses is dominated by the length of the anchorage of the fibers in the matrix and much less by their number. For obvious reasons, in order to maximize the ability of the fibers to transmit tensile stresses, they must be firmly anchored.

### 4.3. Effect of the Matrix Type on the Improvement Effect Achieved

The effectiveness of the dispersed reinforcement is, as already described, strongly dependent on the mechanical parameters of the reinforced concrete. This effect is presented in Figure 16, which compares load—CMOD relationship graphs for BCFPF and SFPF fibers in 400/1.6 and 450/0.8 matrices. Fibers with a length of 60 mm and dosage rate of 10 kg/m^3^ were compared. The 400/1.6 matrix is characterized by the lowest strength and thus the expected lowest fiber adhesion to the matrix. Therefore, the use of a large number of fibers, but short ones, does not guarantee high σ_MOR_ values. On the other hand, in a matrix with high strength (450/0.8) and high fiber adhesion, the same amount, type, and length of fibers results in very high σ_MOR_ values. The calculated maximum delta of the σ_MOR_ stresses due to the type of matrix in the cases described is as much as 3.6 MPa.

On the other hand, the lowest effect of the matrix on the effectiveness of dispersed reinforcement was observed for BCFPF and SFPF fibers with a length of 100 mm and dosage rate of 6 kg/m^3^. For this comparison, the 450/0.8 and 450/1.2 matrices were selected, i.e., of high and medium strength. Both matrices guaranteed sufficient anchorage strength of the long fibers and the quantity was at a lower dosage level. Such a combination of composites selected for comparison showed that the effect of the matrix is not so important in these cases and the obtained σ_MOR_ values were very similar.

Figure 17 shows the comparison of load—CMOD relationship graphs for composites with 450/0.8 and 450/1.2 matrices.

### 4.4. Effect of Dispersed Reinforcement on Total Fracture Energy G_f,tot_

The total fracture energy of the composite G_f,tot_ characterizes the behavior of the composite in the entire range of deflection, regardless of where the characteristic points of the force-deflection relationship occur. In the case of analyzed composites, this parameter proved to be sensitive to variable material parameters used in the testing program. Generally, regardless of the type and quantity of the applied dispersed reinforcement, the values of total fracture energy are strongly dependent on the strength of the matrix. The smallest, because about 30% increase of G_f,tot_ between the weakest and strongest matrix was recorded for BCFPF/100/6 reinforcement. In turn, using BCFPF/60/6 reinforcement, more than 3.5 times the value of this parameter was observed.

Analyzing the influence of the amount of dispersed reinforcement on the determined values of total fracture energy G_f,tot_, it can be concluded that the increase of the fiber dosage level from 6 to 10 kg/m^3^ translated into the expected increase of G_f,tot_ by 46% on average for composites with the 400/1.6 matrix and by 51% with the 450/0.8 matrix. In the case of fiber-reinforced soil-cements with the 450/1.2 matrix, the lowest effect of fiber quantity change was observed on the mean level of 12%. The greatest variation was observed in the case of the composite made of 400/1.6 matrix with BCFPF/60 reinforcement, where the increase in total fracture energy was more than twofold.

Considering the effect of the change in fiber length from 60 to 100 mm on the determined G_f,tot_ values, it was observed that for composites with the 400/1.6 and 450/1.2 matrices, this procedure resulted in mean increases in G_f,tot_ values by 54% and 24%. In the case of fiber-reinforced soil-cement with a 450/0.8 matrix, on the other hand, the increase in fiber length resulted in a 12% reduction in the mean G_f,tot_ value.

## 5. Conclusions

The use of dispersed reinforcement in the form of plastic fibers increases the flexural tensile strength of soil-cement composites, obtaining a pseudo-plastic character of damage and gives the possibility of their operation in a post-critical state. The range of variability of both the type, length, and quantity of the dispersed reinforcement described in the article allowed to increase the flexural tensile strength described by the σ_MOR_/σ_LOP_ ratio, approx. twice on average, regardless of the matrix version used. The change from a brittle matrix cracking to a pseudo-plastic character of the composite fracture is visible in the level of recorded post-critical stresses. While in the absence of reinforcement, even at a fraction of a millimeter from the resulting crack, the material is completely destroyed, the presence of appropriately selected dispersed reinforcement ensures that the load is transferred through the material until crack opening values of even 10 mm at the level of stress destroying the composite matrix.

The effectiveness of the improvement obtained depends primarily on the number and length of fibers used. In most cases, the use of fibers in the amount of 10 kg/m^3^ allowed to obtain higher values of determined mechanical parameters in comparison with a 6 kg/m^3^ dosing level. In general, a similar trend can be observed for 100 mm fibers compared to 60 mm fibers. However, in this case, it applies composites with 400/1.6 and 450/1.2 matrices. For fiber-reinforced soil-cements with the strongest 450/0.8 matrix, higher values of the mechanical properties determined were achieved by specimens with 60 mm-long fibers. In this matrix, the length of 60 mm proved to be sufficient to ensure an adequate level of adhesion of the fibers, and the dominant parameter determining the σ_MOR_ stress value was the amount of reinforcement in the composition.

The presented research results show the influence of the type of soil-cement matrix on the efficiency of the dispersed reinforcement in fiber-reinforced soil-cements. The stronger the soil-cement matrix, the better the effectiveness of the inclusions. The effect of the matrix type on the flexural tensile strength achieved by fiber-reinforced soil-cement is most evident when using 60 mm (shorter) fibers at the dose of 10 kg/m^3^.

The global effect of the analyzed variable material parameters on the obtained results of strengthening of the soil-cement matrix can be observed following an analysis of the values of total fracture energy G_f,tot_. This parameter proved to be sensitive both to the amounts and length of fibers and to the variable matrix strength of the composites tested.

## Figures and Tables

**Figure 1 materials-13-03917-f001:**
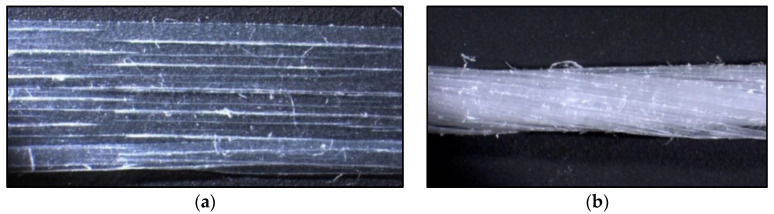
Image of the fibers selected for testing, 10× magnification: (**a**) single fibrillated-tapes polypropylene fiber SFPF (single fibrillated-tapes polypropylene fibers) (400 tex); (**b**) bundle of coiled fibrillated-tapes polypropylene fiber BCFPF (bundles of coiled fibrillated-tapes polypropylene fibers) (2000 tex).

**Figure 2 materials-13-03917-f002:**
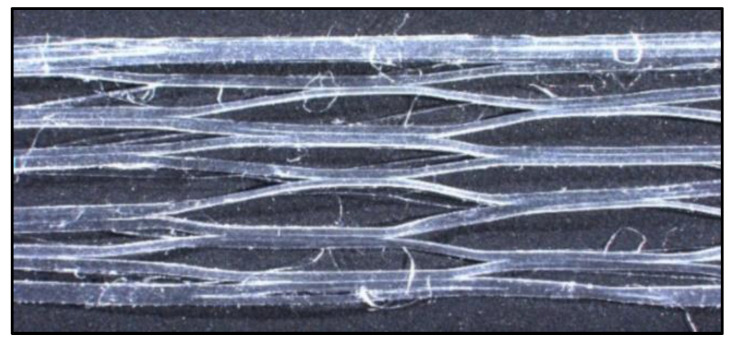
Image of SFPF (single fibrillated-tapes polypropylene fibers) fiber stretched transversely as a result of mixing the components of soil-cement, 10× magnification.

**Figure 3 materials-13-03917-f003:**
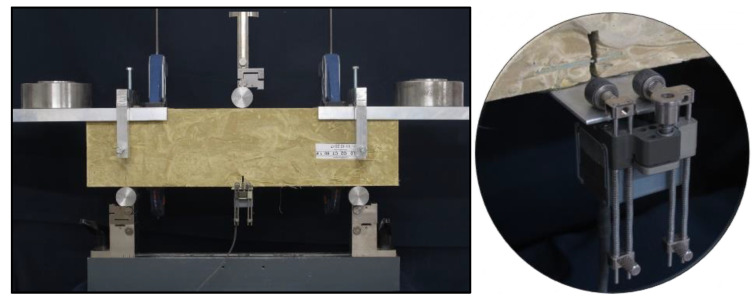
Specimen viewed from the side during the test—CMOD (crack mouth opening displacement) measured using a precision extensometer.

**Figure 4 materials-13-03917-f004:**
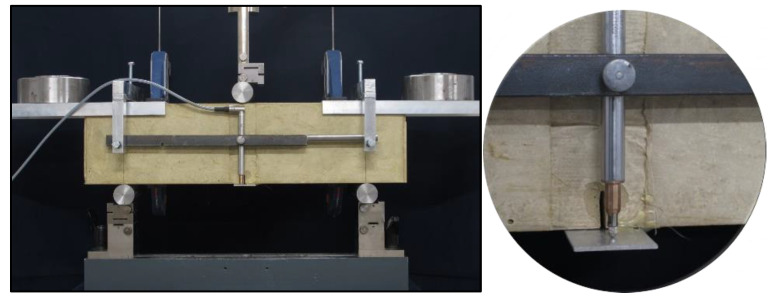
Specimen viewed from the other side during the test—deflection d measured using a displacement transducer.

**Figure 5 materials-13-03917-f005:**
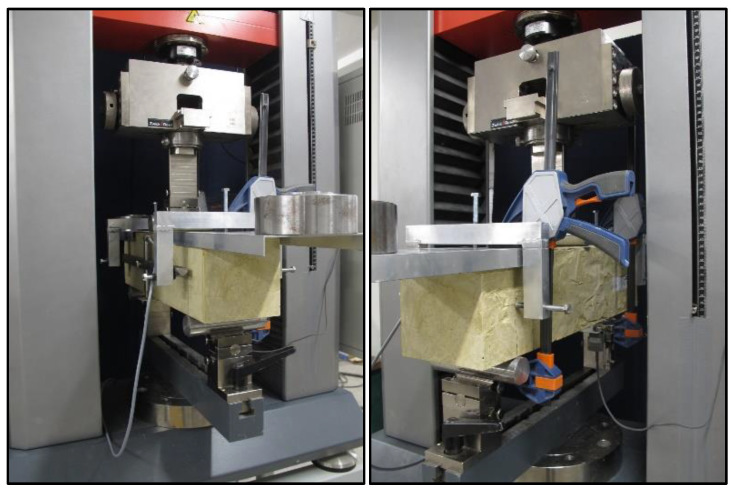
Specimen in a universal testing machine with the frame, counterweights, and sensors.

**Figure 6 materials-13-03917-f006:**
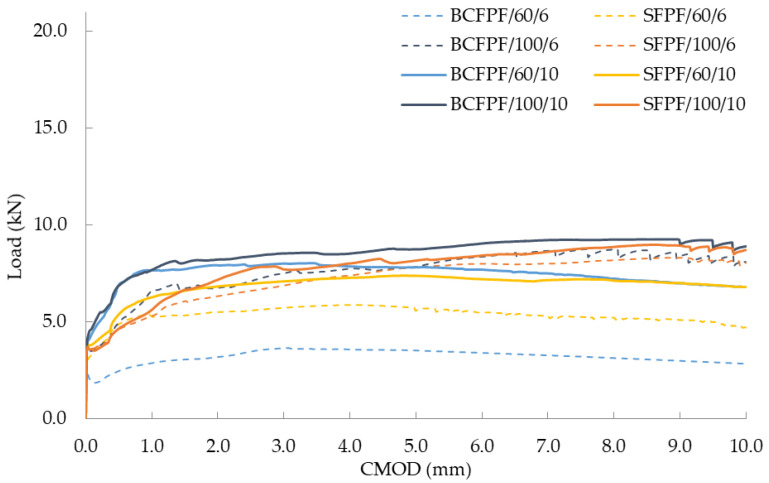
Load—CMOD (crack mouth opening displacement) relationship graphs for 400/1.6 matrix after 28 days of curing.

**Figure 7 materials-13-03917-f007:**
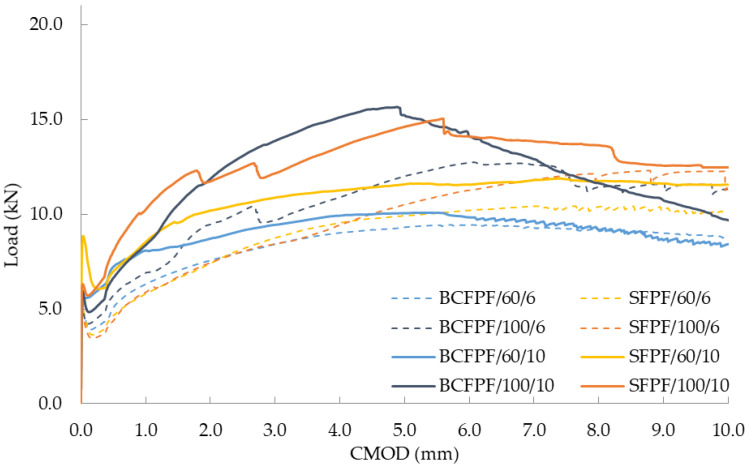
Load—CMOD (crack mouth opening displacement) relationship graphs for 450/1.2 matrix after 28 days of curing.

**Figure 8 materials-13-03917-f008:**
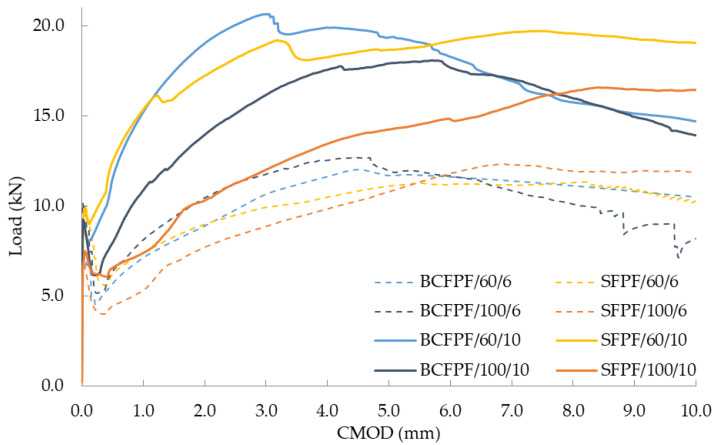
Load—CMOD (crack mouth opening displacement) relationship graphs for 450/0.8 matrix after 28 days of curing.

**Figure 9 materials-13-03917-f009:**
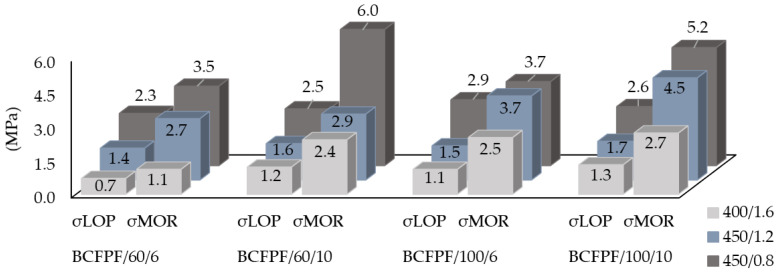
Graphs presenting a combination of σ_LOP_ and σ_MOR_ stress values for fiber-reinforced soil-cements with the addition of BCFPF (bundles of coiled fibrillated-tapes polypropylene fibers).

**Figure 10 materials-13-03917-f010:**
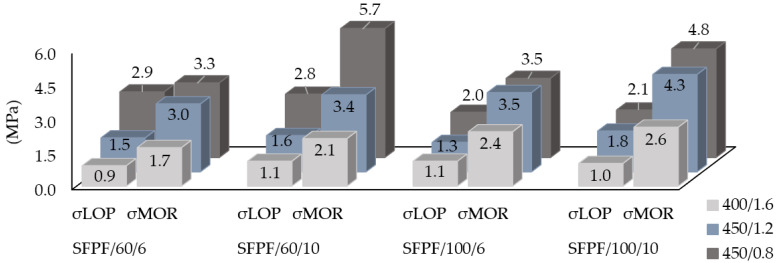
Graphs presenting a combination of σ_LOP_ and σ_MOR_ stress values for fiber-reinforced soil-cements with the addition of SFPF (single fibrillated-tapes polypropylene fibers).

**Figure 11 materials-13-03917-f011:**
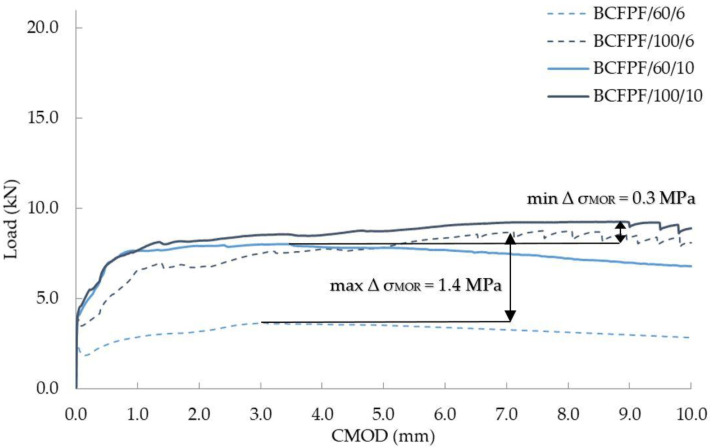
Load—CMOD (crack mouth opening displacement) relationship graphs showing the influence of fiber length on the obtained σ_MOR_ stress values for fiber-reinforced soil-cement with a 400/1.6 matrix.

**Figure 12 materials-13-03917-f012:**
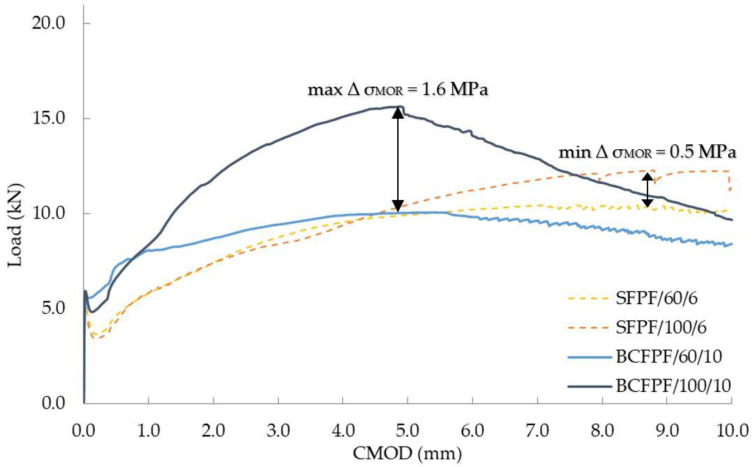
Load—CMOD (crack mouth opening displacement) relationship graphs showing the influence of fiber length on the obtained σ_MOR_ stress values for fiber-reinforced soil-cement with a 450/1.2 matrix.

**Figure 13 materials-13-03917-f013:**
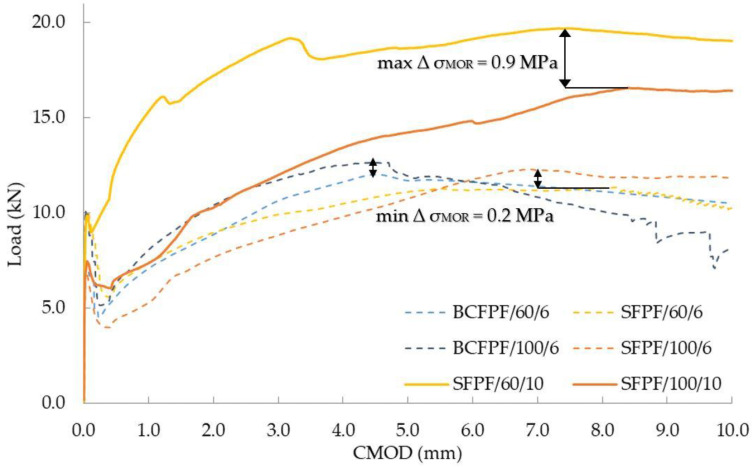
Load—CMOD (crack mouth opening displacement) relationship graphs showing the influence of fiber length on the obtained σ_MOR_ stress values for fiber-reinforced soil-cement with a 450/0.8 matrix.

**Figure 14 materials-13-03917-f014:**
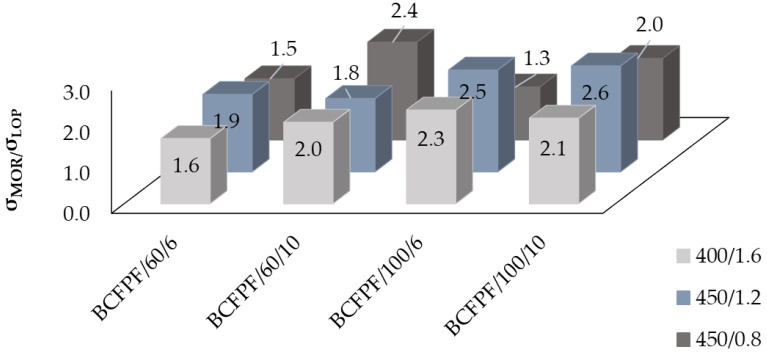
Graphs of the σ_MOR_/σ_LOP_ ratio for BCFPF (bundles of coiled fibrillated-tapes polypropylene fibers) depending on the length and amount of fibers.

**Figure 15 materials-13-03917-f015:**
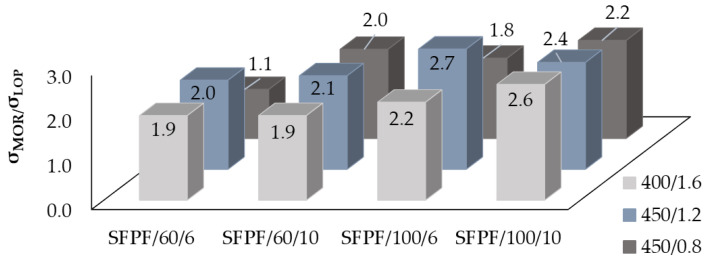
Graphs of the σ_MOR_/σ_LOP_ ratio for SFPF (single fibrillated-tapes polypropylene fibers) depending on the length and amount of fibers.

**Figure 16 materials-13-03917-f016:**
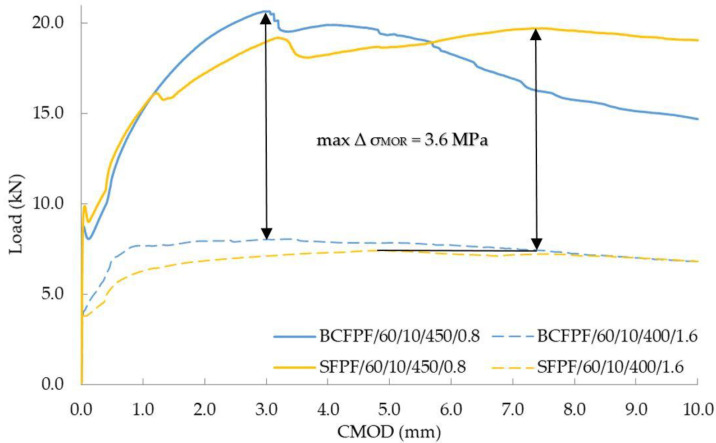
Load—CMOD (crack mouth opening displacement) relationship graphs for fiber-reinforced soil-cements with maximum σ_MOR_ differential stress values.

**Figure 17 materials-13-03917-f017:**
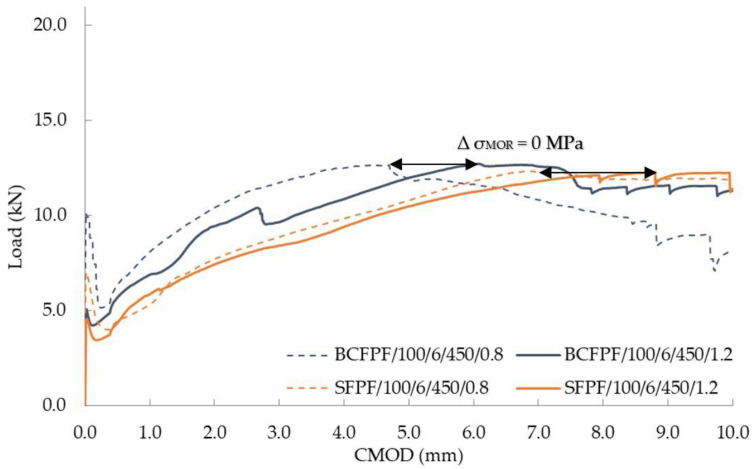
Load—CMOD (crack mouth opening displacement) relationship graphs for fiber-reinforced soil-cements with no σ_MOR_ differential stress values.

**Table 1 materials-13-03917-t001:** Basic properties of polypropylene fibers [8].

Diameter (μm)	Specific Gravity (g/cm^3^)	Modulus of Elasticity (GPa)	Tensile Strength (MPa)	Length Range (mm)	Range of Fiber Amount Used (% by Mass)
23–150	0.92	3–3.5	120–450	6–50	0–3

**Table 2 materials-13-03917-t002:** Basic properties of non-cohesive soil selected for testing.

Properties	Medium Sand Values
Particle size distribution (% by mass) – gravel fractions 2/40 mm – sand fractions 0.05/2 mm – dust fractions 0.002/0.05 mm – loam fractions < 0.002 mm	5 95 - -
Bulk density ρ (g/cm^3^)	1.60
Bulk density of the soil skeleton ρ_d_ (g/cm^3^)	1.56
Maximum bulk density of the soil skeleton ρ_dmax_ (g/cm^3^)	1.86
Natural moisture content w_n śr_ (%)	2.76
Optimal moisture content w_opt_ (%)	8.5
Yield point w_P_ (%)	-
Liquid limit w_L_ (%)	-
Plasticity index I_L_ (%)	-
Approximate content of CaCO_3_ (%)	<1

**Table 3 materials-13-03917-t003:** Basic properties of CEM II/B-S 32.5 R cement.

Properties	Values
Compressive strength after 2 days (MPa)	18.0
Compressive strength after 28 days (MPa)	49.0
Initial setting time (min)	190
Water at normalized consistency (%)	28
Blaine surface area (cm^2^/g)	3570

**Table 4 materials-13-03917-t004:** Basic properties of three soil-cement matrices selected for testing.

Matrix Designation	Paste Content V_z_ (dm^3^/m^3^)	w/c Ratio	Cement Content (kg/m^3^)	Flexural Strength after 28 Days (MPa)	Compressive Strength after 28 Days (MPa)
400/1.6	400	1.6	208	0.9	3.6
450/1.2	450	1.2	296	1.5	8.9
450/0.8	450	0.8	401	2.4	21.4

**Table 5 materials-13-03917-t005:** Mechanical parameters of fiber-reinforced soil-cement with 400/1.6 matrix after 28 days of curing.

Type of Fiber	Fiber Mass Fraction (kg/m^3^)	f_cm_ (MPa)	σ_LOP_ (MPa)	σ_MOR_ (MPa)	σ_MOR/_σ_LOP_	G_f,tot_ (kJ/m^2^)	σ_CMOD_ (MPa)				
							0.5	2.5	5.0	7.5	10.0
BCFPF/60	6	3.2	0.7	1.1	1.6	1.70	0.7	1.0	1.0	0.9	0.8
BCFPF/100		5.9	1.1	2.5	2.3	4.15	1.4	2.0	2.3	2.5	2.3
SFPF/60		4.7	0.9	1.7	1.9	2.74	1.4	1.6	1.6	1.5	1.4
SFPF/100		5.4	1.1	2.4	2.2	3.91	1.3	1.9	2.2	2.3	2.3
BCFPF/60	10	4.6	1.2	2.4	2.0	3.97	2.0	2.3	2.3	2.1	2.0
BCFPF/100		6.2	1.3	2.7	2.1	4.56	2.0	2.4	2.5	2.7	2.6
SFPF/60		3.3	1.1	2.1	1.9	3.61	1.6	2.0	2.1	2.1	2.0
SFPF/100		5.2	1.0	2.6	2.6	4.21	1.3	2.2	2.4	2.5	2.5

**Table 6 materials-13-03917-t006:** Mechanical parameters of fiber-reinforced soil-cement with 400/1.2 matrix after 28 days of curing.

Type of Fiber	Fiber Mass Fraction (kg/m^3^)	f_cm_ (MPa)	σ_LOP_ (MPa)	σ_MOR_ (MPa)	σ_MOR/_σ_LOP_	G_f,tot_ (kJ/m^2^)	σ_CMOD_ (MPa)				
							0.5	2.5	5.0	7.5	10.0
BCFPF/60	6	7.9	1.4	2.7	1.9	4.46	1.5	2.3	2.7	2.7	2.5
BCFPF/100		10.0	1.5	3.7	2.5	5.67	1.6	2.9	3.5	3.5	3.3
SFPF/60		7.3	1.5	3.0	2.0	4.91	1.3	2.4	2.9	3.0	2.9
SFPF/100		7.6	1.3	3.5	2.7	5.60	1.3	2.3	3.0	3.5	3.3
BCFPF/60	10	8.9	1.6	2.9	1.8	4.56	2.1	2.6	2.9	2.7	2.4
BCFPF/100		11.0	1.7	4.5	2.6	6.16	1.9	3.8	4.4	3.5	2.8
SFPF/60		8.2	1.6	3.4	2.1	5.73	2.0	3.0	3.3	3.4	3.3
SFPF/100		10.0	1.8	4.3	2.4	6.83	2.3	3.6	4.2	3.9	3.6

**Table 7 materials-13-03917-t007:** Mechanical parameters of fiber-reinforced soil-cement with 400/0.8 matrix after 28 days of curing.

Type of Fiber	Fiber Mass Fraction (kg/m^3^)	f_cm_ (MPa)	σ_LOP_ (MPa)	σ_MOR_ (MPa)	σ_MOR/_σ_LOP_	G_f,tot_ (kJ/m^2^)	σ_CMOD_ (MPa)				
							0.5	2.5	5.0	7.5	10.0
BCFPF/60	6	18.9	2.3	3.5	1.5	6.23	1.6	2.8	3.4	3.2	3.0
BCFPF/100		18.8	2.9	3.7	1.3	5.40	1.8	3.2	3.4	3.0	2.4
SFPF/60		19.5	2.9	3.3	1.1	5.24	1.8	2.7	3.2	3.2	3.0
SFPF/100		16.5	2.0	3.5	1.8	5.50	1.3	2.4	3.1	3.5	3.4
BCFPF/60	10	18.4	2.5	6.0	2.4	8.79	3.3	5.8	5.6	4.7	4.2
BCFPF/100		18.8	2.6	5.2	2.0	7.63	2.3	4.4	5.2	4.8	4.0
SFPF/60		17.3	2.8	5.7	2.0	9.69	3.6	5.2	5.4	5.7	5.5
SFPF/100		15.7	2.1	4.8	2.2	7.45	1.9	3.2	4.1	4.6	4.7

**Table 8 materials-13-03917-t008:** Comparison of σ_MOR_ stress values and the corresponding CMOD values.

Matrix Designation		400/1.6		450/1.2		450/0.8	
Type of Fiber	Fiber Mass Fraction (kg/m^3^)	σ_MOR_ (MPa)	CMOD (mm)	σ_MOR_ (MPa)	CMOD (mm)	σ_MOR_ (MPa)	CMOD (mm)
BCFPF/60	6	1.1	3.1	2.7	6.3	3.5	4.5
BCFPF/100		2.5	7.6	3.7	6.1	3.7	4.5
SFPF/60		1.7	4.0	3.0	7.6	3.3	8.2
SFPF/100		2.4	9.1	3.5	8.8	3.5	6.9
BCFPF/60	10	2.4	3.4	2.9	5.2	6.0	3.0
BCFPF/100		2.7	8.9	4.5	4.9	5.2	5.7
SFPF/60		2.1	4.8	3.4	7.4	5.7	7.3
SFPF/100		2.6	8.6	4.3	5.6	4.8	8.4

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
