# Peer review of "The Effect of the Type and Amount of Synthetic Fibers on the Effectiveness of Dispersed Reinforcement in Soil-Cements"

_materials, 2020, doi:10.3390/ma13183917_

Round 1
Reviewer 1 Report
Abstract:
I don’t know the usefulness of the numbering in bracket in the abstract, please remove, the numbering in brackets (1), (2) and (3) from the abstract.
Line 10:
Delete (1)
Line 13:
Delete (3)
Line 14 should be:
reinforcement: Single Fibrillated-tapes Polypropylene Fibres (SFPF) and Bundles of Coiled Fibrillated-tapes Polypropylene Fibres (BCFPF).
Revised line 17, the sentence is not clear.
Line 18. 19 & 20 should be:
tests were carried out to determine post-critical stress values, the total fracture energy and compressive strength. The test results obtained, and
Introduction
Revised line 72, the sentence is not clear.
Materials and Methods
Line 158, 159 & 160 should be:
Two types of polypropylene fibres were analysed for behaviour together with these matrices:
Single Fibrillated-tapes Polypropylene Fibres (SFPF) and Bundles of Coiled Fibrillated-tapes Polypropylene Fibres (BCFPF), which differ in terms of length – 60 and 100 mm, for details see Figure 1. The choice of the length of fibres
Lines 237 & 238 should be:
and possible displacement of the supports during the test). The details regarding the test stand can be seen in Figure 5.
Results and discussion
Line 248 should be:
presented in Figures 6–8. In the Tables 5 and Figure 6, the markings containing information about:
Lines 324 should be:
Since the σMOR stress value was recorded at different values of CMOD, the Table 8 below shows
Author Response
Response to Reviewer 1 Comments
First of all, the authors wishes to thank you very much for your valuable time spent to prepare this review, and express gratitude for valuable comments. In the light of those comments the minor changes have been made to the article.
Point 1: I don’t know the usefulness of the numbering in bracket in the abstract, please remove, the numbering in brackets (1), (2) and (3) from the abstract.
Response 1: In accordance with your guidance, we have made changes in the article.
Point 2: Line 14 should be: reinforcement: Single Fibrillated-tapes Polypropylene Fibres (SFPF) and Bundles of Coiled Fibrillated-tapes Polypropylene Fibres (BCFPF).
Response 2: In accordance with your guidance, we have made changes in the article.
Point 3: Revised line 17, the sentence is not clear.
Response 3: In accordance with your guidance, we have made changes in the article.
Point 4: Line 18. 19 & 20 should be: tests were carried out to determine post-critical stress values, the total fracture energy and compressive strength. The test results obtained, and
Response 4: It is important for us to mention σLOP and σMOR in the abstract as these are key parameters for the evaluation of matrix reinforcement with fibres.
Point 5: Revised line 72, the sentence is not clear
Response 5: In accordance with your guidance, we have made changes in the article.
Point 6: Line 158, 159 & 160 should be:
Two types of polypropylene fibres were analysed for behaviour together with these matrices:
Single Fibrillated-tapes Polypropylene Fibres (SFPF) and Bundles of Coiled Fibrillated-tapes Polypropylene Fibres (BCFPF), which differ in terms of length – 60 and 100 mm, for details see Figure 1. The choice of the length of fibres
Response 6: In accordance with your guidance, we have made changes in the article.
Point 7: Lines 237 & 238 should be:
and possible displacement of the supports during the test). The details regarding the test stand can be seen in Figure 5
Response 7: In accordance with your guidance, we have made changes in the article.
Point 8: Line 248 should be:
presented in Figures 6–8. In the Tables 5 and Figure 6, the markings containing information about:
Response 8: In accordance with your guidance, we have made changes in the article.
Point 9: Lines 324 should be:
Since the σMOR stress value was recorded at different values of CMOD, the Table 8 below shows
Response 9: In accordance with your guidance, we have made changes in the article.

Reviewer 2 Report
The manuscript presents te results of experimental study with discussion on the effect of application of disperced reinforcement with synthetic fibres in soil-cement material.
The topic of the paper is interesting and valuable. It is well written, with thorough and logical theoretical background and overview of the subject, properly designed methods and clrearily presented results. The results are discussed and exaplained, and conclusions result naturally from these observations.
I recommend publication of this paper, however, I would like to ask the Authors for minor improvements (mainly addition of some more detailed explanations of the methodological assumptions) as follows:
- Could the Authors exaplin at the begining of the objective of the study why they chose to investigate the influence of sythetic dispersed reinforcement? As stated in the introduction, non-metallic reinforcement impacts mostly shrinkage of cement-based matrix and improves ductile behaviour of the composite, but does not influence its strength as effectively as metallic ones. What was the primary objective of the Authors? Could you make a comment on the environmental impact of such reinforcement being introduced into the subgrade?
- Page 4, line 143-144 - please explain why the Authors assumed the basic criterion to limit the cement content to the specified values? The Authors state that the ready mix should exhibit a specific liquidity but was the aim to have as little or as much cement as possible?
- Page 5, line 175 - also please explain the choice for the fibres content. In case of fibre-reinforced concrete this content is usually given as a percentage of volume, is there a correlation?
- Also, the Authors make other comparisons between fibre-reinforced concrete and soil-cement. I think that a brief comparison of the two materials at the begining of the paper could be helpful in general for understainding of the assumptions made by the Authors regarding the mix design or the choice of methods. This comparison could include mix composition (especially regarding the typically used type and amount of cement and amount of fine aggregate) as well as basic mechanical paramters (strengths, stiffness).
- Page 4 line 151, Page 6, line 197; Page 8, line 240 - whenever the Authors refer to a standard, please give its name already in the text (it will be easier to read then to go to the References).
- I suggest that a brief discussion of the standards chosen for the experimental tests is presented already on Page 6, line 200 - this would explain the Authors' choice. Now the Eurocode is discussed first, and then the RILEM report. Only after that the reader gets the full understanding on the methods given by the two and their complementary character.
- Page 14, line 378-380 - please rephrase the sentence (there is an idem per idem mistake): "[...] it appears that in matrices with a lower strength, the length of the anchorage of the fibres in the matrix is dominated by the length of the anchorage of the fibres in the matrix and much less by their number."
Author Response
Response to Reviewer 2 Comments
First of all, the authors wishes to thank you very much for your valuable time spent to prepare this review, and express gratitude for valuable comments. In the light of those comments the minor changes have been made to the article.
Point 1: Could the Authors exaplin at the begining of the objective of the study why they chose to investigate the influence of sythetic dispersed reinforcement? As stated in the introduction, non-metallic reinforcement impacts mostly shrinkage of cement-based matrix and improves ductile behaviour of the composite, but does not influence its strength as effectively as metallic ones. What was the primary objective of the Authors? Could you make a comment on the environmental impact of such reinforcement being introduced into the subgrade?
Response 1: Thank you for your careful guidance and help. In accordance with your guidance, we have made changes in the article.
The selected synthetic dispersed reinforcement is resistant to the acidic environment, which may be soil used to made soil-cement composites.
Point 2: Page 4, line 143-144 - please explain why the Authors assumed the basic criterion to limit the cement content to the specified values? The Authors state that the ready mix should exhibit a specific liquidity but was the aim to have as little or as much cement as possible?
Response 2: Thank you for your careful guidance and help. In accordance with your guidance, we have made changes in the article.
Point 3: Page 5, line 175 - also please explain the choice for the fibres content. In case of fibre-reinforced concrete this content is usually given as a percentage of volume, is there a correlation?
Response 3: The fibre content is given in (kg/m3) due to the ease of use in geotechnical practice. The adopted levels of fibre dosage result from preliminary tests. The fibre content less than 6 kg/m3 doesn’t make it possible to obtain a satisfactory level of soil-cement matrix reinforcement. The use of fibres in an amount greater than 10 kg/m3 would affect the uniformity of fibres dispersion in soil-cement matrix.
Point 4: Also, the Authors make other comparisons between fibre-reinforced concrete and soil-cement. I think that a brief comparison of the two materials at the begining of the paper could be helpful in general for understainding of the assumptions made by the Authors regarding the mix design or the choice of methods. This comparison could include mix composition (especially regarding the typically used type and amount of cement and amount of fine aggregate) as well as basic mechanical paramters (strengths, stiffness).
Response 4: Certainly, these two composites are very different in terms of their compositions. On the one hand, in the case of concrete, we have aggregate. On the other hand, in the case of soil-cement, there is native soil instead of aggregate. We can also mention different amounts of cement, different w/c ratio values or different amounts and types of added fibres. Therefore, the obtained properties of both materials are very different and we believe that the comparison of these two materials is unfounded. In the article we focus on fibre-reinforced soil-cement and not on other types of cement composites.
Point 5: Page 4 line 151, Page 6, line 197; Page 8, line 240 - whenever the Authors refer to a standard, please give its name already in the text (it will be easier to read then to go to the References).
Response 5: Thank you for your careful guidance and help. In accordance with your guidance, we have made changes in the article.
Point 6: I suggest that a brief discussion of the standards chosen for the experimental tests is presented already on Page 6, line 200 - this would explain the Authors' choice. Now the Eurocode is discussed first, and then the RILEM report. Only after that the reader gets the full understanding on the methods given by the two and their complementary character.
Response 6: Thank you for your careful guidance and help. In accordance with your guidance, we have made changes in the article. Page 6, line 203.
Point 7: Page 14, line 378-380 - please rephrase the sentence (there is an idem per idem mistake): "[...] it appears that in matrices with a lower strength, the length of the anchorage of the fibres in the matrix is dominated by the length of the anchorage of the fibres in the matrix and much less by their number."
Response 7: Thank you for your careful guidance and help. In accordance with your guidance, we have made changes in the article.
